# Construction of the Global Reference Atmospheric Profile Database

Yuhang Guo [1,2,3], Xiaoying Li [1,*], Tianhai Cheng [1] , Shenshen Li [1], Xinyuan Zhang [1,2,3], Wenjing Lu [1,2,3] and Weifang Fang [4]

1 Aerospace Information Research Institute, Chinese Academy of Sciences, Beijing 100094, China; guoyuhang21@mails.ucas.ac.cn (Y.G.); chength@radi.ac.cn (T.C.); lishenshen@aircas.ac.cn (S.L.); zhangxinyuan20@mails.ucas.edu.cn (X.Z.); luwenjing22@mails.ucas.ac.cn (W.L.)
2 University of Chinese Academy of Sciences, Beijing 100101, China
3 College of Resources and Environment, University of Chinese Academy of Sciences, Beijing 101408, China
4 College of Resources and Environment, Hefei University of Technology, Hefei 230009, China; fang011006@163.com
* Correspondence: lixy01@radi.ac.cn

**Abstract:** Atmospheric profiles are important input parameters for atmospheric radiative transfer models and atmospheric parameter inversions. The construction of regionally representative reference atmospheric profiles can provide basic data for global atmospheric and environmental research. Most reference atmospheric profile databases commonly used lag behind in updating frequency. These databases usually have limited spatial and temporal resolution and differ greatly from the real atmospheric state. To present the real atmospheric state, this article constructs the Global Reference Atmospheric Profile Database (GRAP) based on ACE-FTS satellite products of 2021 and 2022, AIRS satellite products and ERA5 reanalysis data of 2022 u6sing a random forest regression model and a hierarchical mean algorithm. The radiance spectrum of FY-3E HIRAS-II using different profile databases was simulated and compared with the measured spectrum. The results show that GRAP spectral simulations fit better with the measured HIRAS-II spectrum. Comparing the $CO_2$, $CH_4$, $O_3$ and $N_2O$ profiles of GRAP, AFGL, MIPAS, RTTOV and NDACC ground station profiles in equatorial, mid-latitude summer and polar winter, the results show that GRAP has high spatial and temporal resolution and better fits the current real atmospheric state. Comparing the temperature profiles of eight regions in China, the results illustrate that GRAP is a better representation of the state of the atmosphere in the Chinese region. GRAP can provide fundamental atmospheric data for radiative transfer studies and atmospheric parameter inversions.

**Keywords:** atmospheric profiles; global reference atmospheric profile database; AIRS; ERA5; ACE-FTS; AFGL; MIPAS; RTTOV

## 1. Introduction

Atmospheric profiles describe the state of the atmosphere and fundamentally determine its optical properties. Atmospheric profile sample datasets and reference atmospheric profile databases are widely used in research on atmospheric radiative transfer models, atmospheric parameter inversions, simulations of the spectral properties of new satellite instruments and satellite data assimilation [1–5]. As the global atmospheric environment changes, the data on atmospheric profiles used for model development and instrument accuracy verification needs to be continuously updated. Therefore, the construction of a regionally representative global reference atmospheric profile database is of significant importance for the atmospheric environment and global change research.

Atmospheric profile sample datasets can be used to estimate the statistical properties of the background fields. Currently, there are several versions of atmospheric profile sample datasets commonly used internationally, such as TIGR (Thermodynamic Initial

Guess Retrieval) [6], ECMWF (European Centre for Medium-Range Weather Forecast) 31L-SD, 50L-SD [6,7], 60L-SD [8–10], NESS-35 [11,12], NOAA88 [13,14], etc. Each sample dataset contains different atmospheric profile parameters, data sources and sample sizes due to their different intended applications. The TIGR is an atmospheric profile sample dataset created by the French Laboratory for Dynamical Meteorology, and there are currently five versions available. These profiles were selected from a large number of atmospheric samples from different periods around the world using topological methods. The ECMWF used the same methods as the TIGR and created the 31L-SD, 50L-SD, 60L-SD and 91-L short-range forecast atmospheric profile sample datasets. The ECMWF creates the NOAA88 atmospheric profile sample dataset of 7547 sounding profiles and the ECMWF-52 sample dataset with 52 atmospheric temperatures, humidity and ozone profiles in two atmospheric height level formats, 60 and 101 layers. The existing studies have analyzed these atmospheric sample datasets and found that only the TIGR-43 sample dataset contains one atmospheric profile located on Dachen Island, Zhejiang Province, China [15]. The other atmospheric sample datasets generally lack atmospheric samples that are representative of the Chinese region. To address this issue, Qi Chengli used the topological sampling method to establish the CRASD-1 and CRASD-2 sample datasets with characteristics specific to the Chinese region [15,16]. Due to the large latitude span, complex topography and diverse climate of China, using a single atmospheric profile to represent the whole Chinese region is unreasonable and may cause significant errors in research and analysis. Therefore, it is crucial to improve the Chinese regional atmospheric profiles.

The reference atmospheric profile databases are primarily used for the application performance evaluation and accuracy verification of satellite detectors, radiative transfer models and atmospheric inversion method models. The databases should include meteorological parameters such as air pressure, temperature, gas composition and profile distribution. For internationally popular atmospheric radiative transfer software, LOWTRAN [17], MODTRAN [18], LBLRTM [19], FASCODE [20] and RFM [21] are used in the six reference atmospheric profiles created by the US Air Force Geophysical Laboratory (AFGL), which are the tropical (15°N) atmosphere, the mid-latitude summer (45°N, July) atmosphere, the mid-latitude winter (45°N, January) atmosphere, the sub-polar summer (60°N, July) atmosphere, the sub-polar winter (60°N, January) atmosphere, and the 1976 US Standard Atmosphere [22,23]. The six reference atmospheric profiles take into account the changes of atmospheric parameters with latitude and season, but their spatial–temporal distribution only represents the summer and winter of latitude zones without considering the influence of longitude on atmospheric parameters or the seasonal changes in spring and autumn. Furthermore, the reference atmospheric profiles are updated less frequently. With the intensifying global climate change, atmospheric parameters such as global temperature, $CO_2$, $CH_4$ and $O_3$ have undergone significant changes compared to previous ones, and the delay in updates could cause great errors in the application of studies applying these reference atmospheric profiles.

To address the issues of the existing reference atmospheric profile databases, such as long update periods, large spatial resolution and inadequate consideration of seasonal changes in spring and autumn, this article uses the ACE-FTS Level 2 Version 4.1 products in 2021 and 2022, the AIRS Support Level 2 Version 7 products, and ERA5 reanalysis data in 2022 to create the Global Reference Atmosphere Profile Database (GRAP) through the use of a random forest regression model and a stratified mean algorithm. The objective is to provide data support for research on global climate change and atmospheric component inversion.

## 2. Materials and Methods

### 2.1. Data Sources

The data sources used in this article include ACE-FTS Level 2 Version 4.1 satellite products in 2021 and 2022, AIRS Support Level 2 Version 7 satellite products, and ERA5 reanalysis data in 2022.

(1) ACE-FTS L2 products

The ACE-FTS instrument was launched on 12 August 2003 on board the SciSat-1 satellite. It has a spectral resolution of $0.02 \text{ cm}^{-1}$, a vertical resolution of 1–2 km, a horizontal resolution of 500 km, a wavelength range of $750–4400 \text{ cm}^{-1}$ (2.2–13.3 μm), and a high vertical resolution using occultation for atmospheric sounding [24]. ACE-FTS Level 2 Version 4.1 is a global dataset that includes pressure, temperature, and more than 40 atmospheric constituents such as $CO_2$, $CH_4$, $H_2O$, $O_3$ and $N_2O$ for the period 2004 to 2023.

(2) AIRS L2 products

The Atmospheric Infrared Sounder (AIRS) on the EOS Aqua Spacecraft was launched on 4 May 2002. It orbits from one pole of the Earth to the other about fifteen times a day, covering the same region of the Earth twice a day. AIRS detects wavelengths in the range $650–2700 \text{ cm}^{-1}$ (3.7–15.4 μm) and has a total of 2378 spectral channels [25]. AIRS can detect vertical profiles of atmospheric temperature and humidity, as well as the trace greenhouse gases $CO_2$, $CH_4$, $O_3$ and CO.

(3) ERA5 reanalysis data

ERA5 reanalysis data is the fifth generation of atmospheric reanalysis products produced by ECMWF, providing hourly data and monthly averages for many atmospheric, land surface and sea state parameters. ERA5 reanalysis data covers the time period from 1940 to the present, with daily ERA5 data updates currently 5 days behind real time. The data is stored in a globally gridded data format, GRIB and NetCDF, with a spatial resolution of $0.25° \times 0.25°$, vertical coverage from 1000 hPa to 1 hPa and a vertical resolution of 37 pressure layers [26].

*2.2. Methods*

In this article, considering the influence of time and space on the atmospheric state, GRAP is divided into January to December according to the month, and the globe is divided into 38 latitude zones and 14 longitude zones, each spanning 5° in latitude and 30° in longitude (2.5° for 0n, 0s, 90n, 90s and 15° for 0e, 0w, 180e, 180w). This division results in a total of 532 grids. GRAP includes two atmospheric state parameters as well as 59 atmospheric component parameters, which are detailed in Table 1. The flow chart of the methods of creating GRAP is shown in Figure 1.

**Table 1.** Metadata information for the GRAP.

| Keywords | Detailed Information |
|---|---|
| Name | Global Reference Atmospheric Profile Database, GRAP |
| Data storage format | .txt |
| Time resolution | January–December, one-month interval (MON1, MON2, MON3, ... , MON11, MON12) |
| Spatial resolution | Latitude zones: 5° interval. n for north, s for south. (0n, 0s, 90n, 90s latitude zones are 2.5°) Longitude zones: 30° interval. e for eastern, w for western. (0e, 0w, 180e, 180w longitude zones are 15°) |
| Height level | 0–119 km at 1km intervals, divided into 120 level |
| Atmospheric parameters | Pressure, Temperature, $CO_2$, $CH_4$, $O_2$, NO, $N_2O$, $O_3$, $SO_2$, $NH_3$, $SF_6$, CO, $N_2$, HF, HBr, $CF_4$, $NO_2$, HI, OCS, $H_2CO$, $H_2O_2$, $C_2H_2$, $C_2H_6$, $PH_3$, $COF_2$, $H_2S$, $CFCl_3$, $CF_2Cl_2$, $CClF_3$, $CHCl_2F$, OH, $CHClF_2$, ClO, $C_2Cl_3F_3$, $C_2Cl_2F_4$, $C_2ClF_5$, $CCl_4$, $ClONO_2$, $N_2O_5$, $HNO_4$, BrO, $CH_3Cl$, $CH_3CN$, $CH_3OH$, $H_2O$, HCl, HCN, $HNO_3$, HCOOH, HOCl, $CCl_2F_2$, $HO_2$, $CCl_3F$, $COCl_2$, COClF, pan($CH_3C(O)OONO_2$), $CHF_3$, $HO_2NO_2$, HCFC141b, HCFC142b, HFC134a |

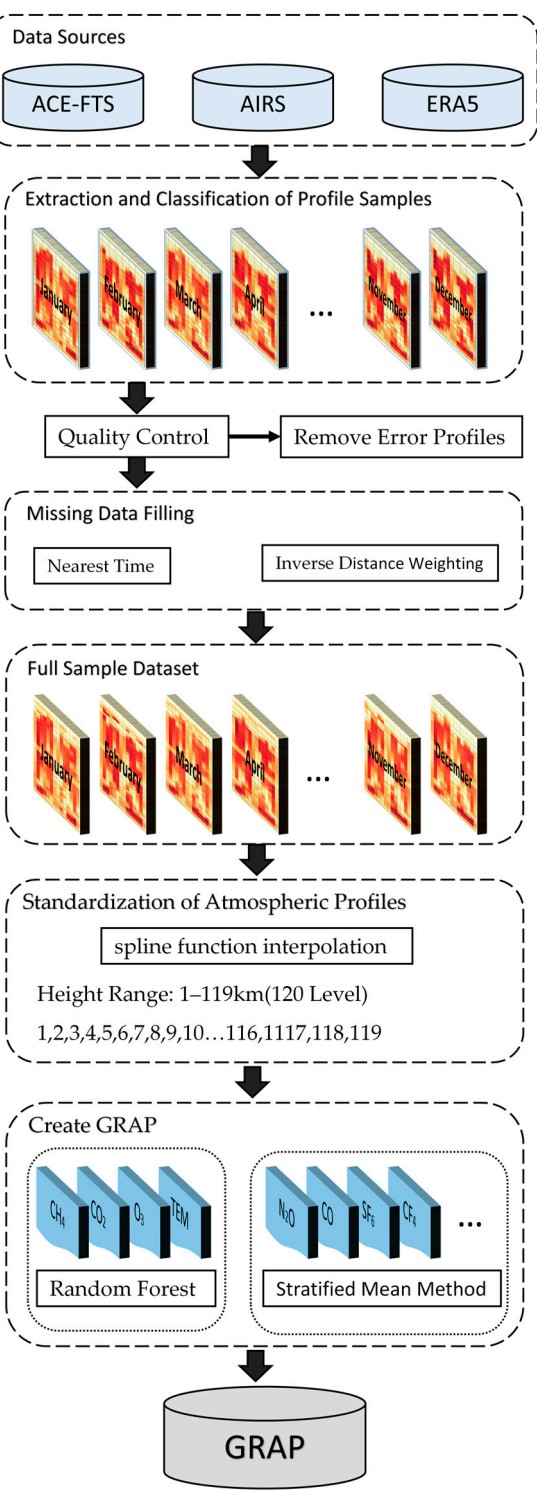

**Figure 1.** Flow chart of the methods of creating GRAP.

### 2.2.1. Database Metadata

Information on the name, data format, spatial and temporal resolution, height stratification format and atmospheric parameters of the global reference atmospheric profile database is in Table 1.

### 2.2.2. Atmospheric Profile Samples Acquisition

The construction of GRAP necessitates meeting the demands for extensive spatial and temporal coverage as well as comprehensive atmospheric composition. Simultaneously, it

is crucial to ensure that the reference atmospheric profiles accurately represent the local atmospheric conditions. The primary challenge lies in extracting realistic profile samples from a vast collection of atmospheric profiles.

Temperature, pressure and volume mixing ratio (vmr) profiles of different atmospheric constituents are derived from the ACE-FTS, AIRS and ERA5 datasets. These profiles are then aligned with a standardized global grid, considering the detection time and latitude/longitude information, resulting in a dataset of 61 atmospheric profiles covering the entire globe. The extracted atmospheric profiles encompass data from all seasons throughout the year for 365 days, providing comprehensive spatial coverage. Figure 2 illustrates the distribution of $CH_4$ profile samples across the global grid, spanning from January to December.

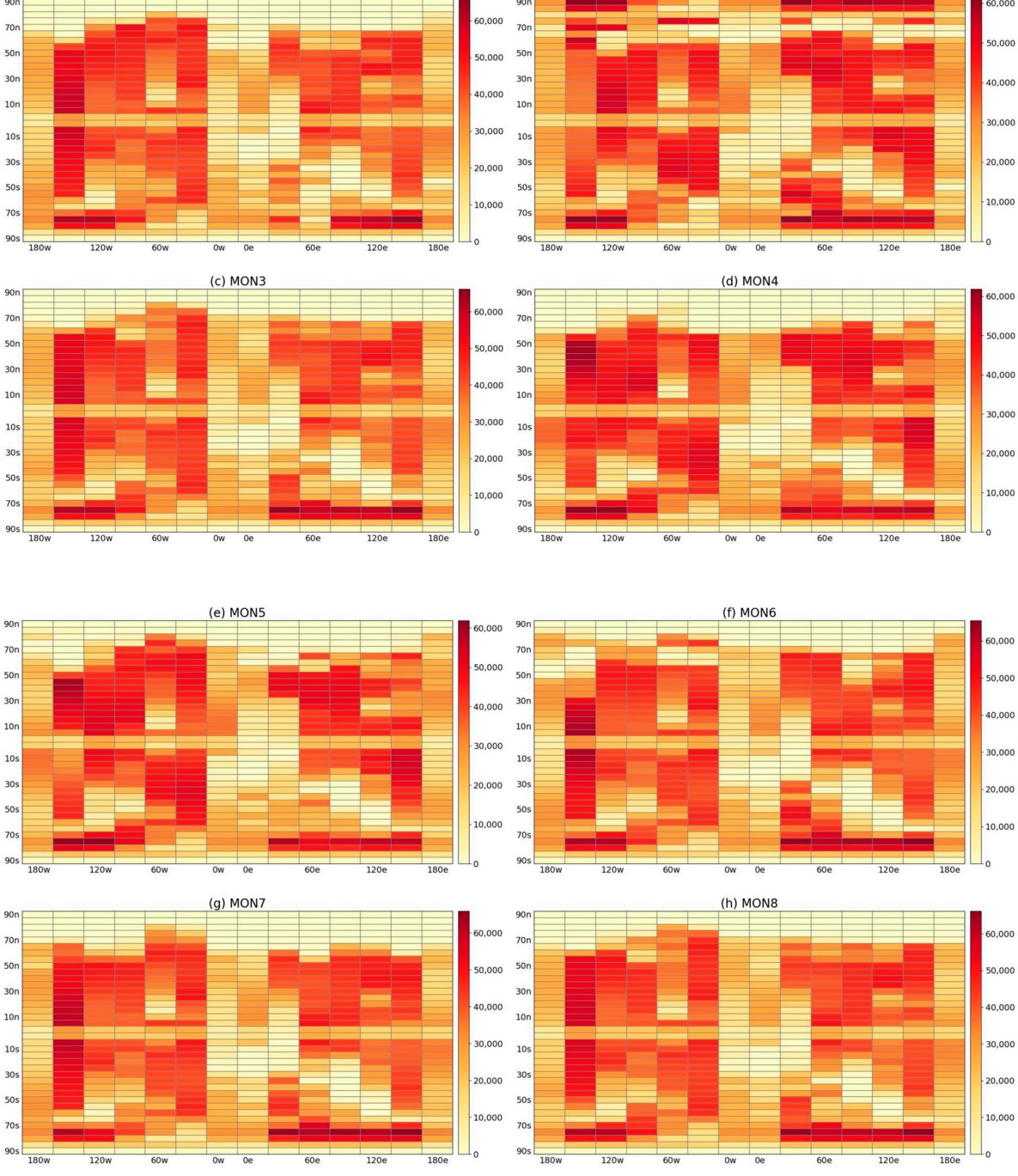

**Figure 2.** *Cont.*

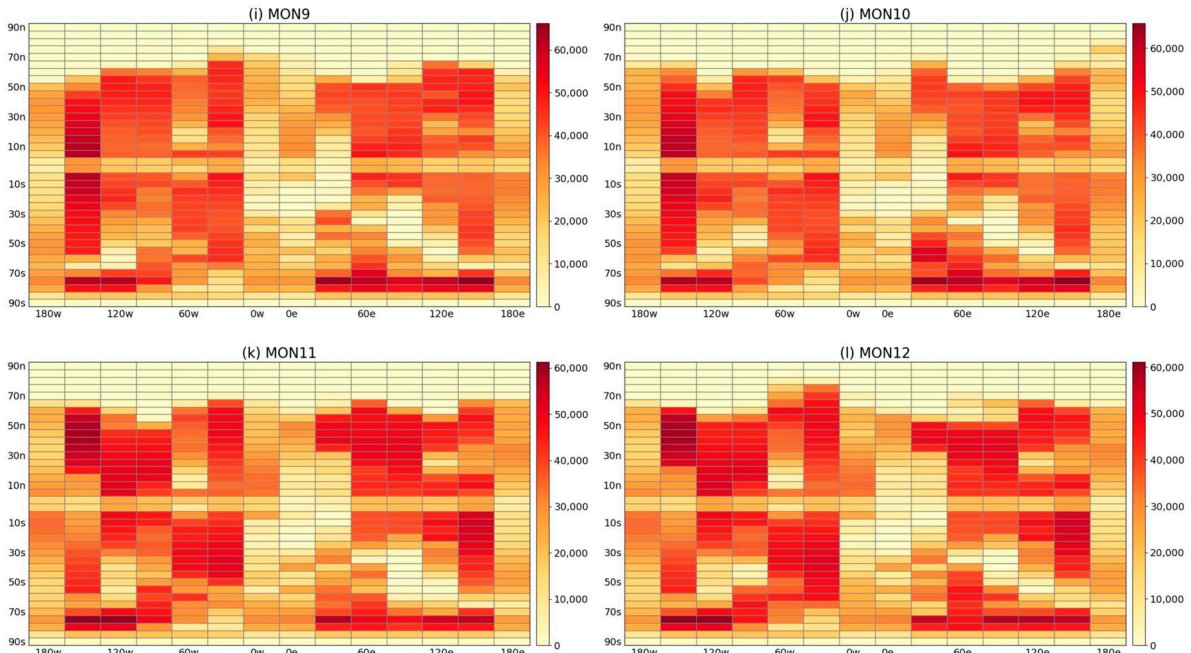

**Figure 2.** Sample numbers of $CH_4$ profiles in the global from January to December.

### 2.2.3. Data Quality Control

To ensure the reliability of the satellite soundings, rigorous quality control measures are implemented to identify and eliminate erroneous profiles caused by detection errors, instrument malfunctions, cloud effects, or other exceptional circumstances. These quality control procedures aim to refine the original profile samples by excluding any incomplete or erroneous values.

The AIRS data comprises the volume mixing ratio (vmr) and the quality control mark (QC) of the atmospheric constituents. QC values are assigned as follows: 0 indicates the highest quality, 1 indicates good quality, and 2 indicates unusable. AIRS data include the vmr and the quality control mark (QC) of the atmospheric constituents. Profiles with a QC mark of 2 were excluded from the AIRS data.

The ACE-FTS data consists of volume mixing ratio (vmr) and vmr errors for different gaseous atmospheric constituents. Compute the error ratio for the volume mixing ratio and eliminate profiles with an error ratio exceeding 15%.

Since temperature, pressure and atmospheric composition vary with height with a certain regularity and the difference between adjacent heights is within a certain range [27], the vertical consistency is used to test the quality of the profiles and use Equation (1) to calculate the rate of vertical change of the atmosphere for each height layer.

$$dx_h = \frac{x_{h+1} - x_h}{H_{h+1} - H_h} \tag{1}$$

where $dx_h$ is the rate of vertical change of the profile in the height layer of $h$, $x_h$ is the profile value in the $h$ layer and $H_h$ is the corresponding height layer.

The standard deviation of the vertical rate of change of the profiles, $\sigma$, as shown in Equation (2), was calculated. Moreover, the error profiles exceeding three $\sigma$ were removed.

$$\sigma = \sqrt{\frac{\sum_{i=1}^{n}(x_i - \mu)^2}{n}} \tag{2}$$

where $x_i$ is the rate of vertical change of the profile in layer $i$ and $\mu$ is the average of $x_i$ across height layers of the profile.

2.2.4. Missing Data Filling

From Figure 1, it can be seen that there are missing atmospheric profiles in some of the grids, resulting in incomplete global data. In this article, the profiles in the missing grids are interpolated using interpolation in the space–time domain to fill the data. Interpolation is performed on the time domain based on the time series, using the data closest in time to the missing data to fill in. In the spatial domain, the inverse distance weight interpolation (IDW) is used as in Equation (3) [28]. using observations around the location of the interpolation point to fill in the missing positions.

$$\hat{Z}_0 = \sum_{i=0}^{n} (Z_i Q_i) \tag{3}$$

where $\hat{Z}_0$ is the estimated value at the point $(x_0, y_0)$, $Q_i$ is the estimated weight coefficient of the interpolated point corresponding to the observed point, and $n$ denotes the number of interpolated points. $Q_i$ is shown in Equation (4) as follows:

$$Q_i = \frac{f(d_{ej})}{\sum_{j=1}^{n} f(d_{ej})} \tag{4}$$

where $n$ is the number of known observation points and $f(d_{ej})$ denotes the weight function of the known distance $d_{ej}$ between the known observation points and the interpolated points. Equation (5) for $f(d_{ej})$ is as follows:

$$f(d_{ej}) = \frac{1}{d_{ej}} \tag{5}$$

2.2.5. Standardization of Atmospheric Profiles

The pressure levels of the different data sources depend on the effective sounding altitude of the instrument. The ACE-FTS data provide an altitude range of 0.5 to 149.5 km, corresponding to a pressure range of 1013 to $3.22 \times 10^{-6}$ hpa, divided into 150 pressure levels. The AIRS data provide a pressure range of 1100 to $1.61 \times 10^{-6}$ hpa, divided into 100 pressure levels. The ERA5 reanalysis data provides a pressure range of 1000 to 1 hPa, divided into 37 pressure levels. In this article, all profile samples are interpolated onto a uniform elevation grid. The profiles have an elevation range of 0–119 km and a vertical interval of 1 km over the entire height range (Table 2 gives the three data sources and the GRAP height range level) [29]. A non-linear relationship between height and sample contour values is constructed in each grid, and each profile is interpolated to a standard height grid using a spline function interpolation method.

**Table 2.** Data sources and GRAP height range level.

| Data Sources | Height Range | Pressure Range | Number of Layers |
|---|---|---|---|
| ACE-FTS | 0.5–149.5 km | 1013–$3.22 \times 10^{-6}$ hpa | 150 |
| AIRS | - | 1100–$1.61 \times 10^{-6}$ hpa | 100 |
| ERA5 | - | 1000–1 hPa | 37 |
| GRAP | 0–119 km | Differences between grids | 120 |

2.2.6. Creation of the Global Reference Atmospheric Profile Database

The data sources for $CH_4$, $CO_2$, $O_3$ and temperature profile samples are mainly from AIRS satellite data and ERA5 reanalysis data, which are large in number. The four atmospheric profile samples in each grid were fitted with a random forest regression model to obtain a standard profile representing that grid. Random Forest (RF) is an algorithm that uses multiple trees to train and predict a sample [30]. There are two advantages to using a random forest regression model:

(1)    The random forest model has a random nature in sample extraction and feature selection, and the algorithm is not prone to over-fitting;

(2)    When creating the random forest, an unbiased estimate of the Generalization Error is used, and the model has a strong generalization capability;

The random forest regression model construction process is as follows. Figure 3 shows the Random Forest Model Construction Flowchart.

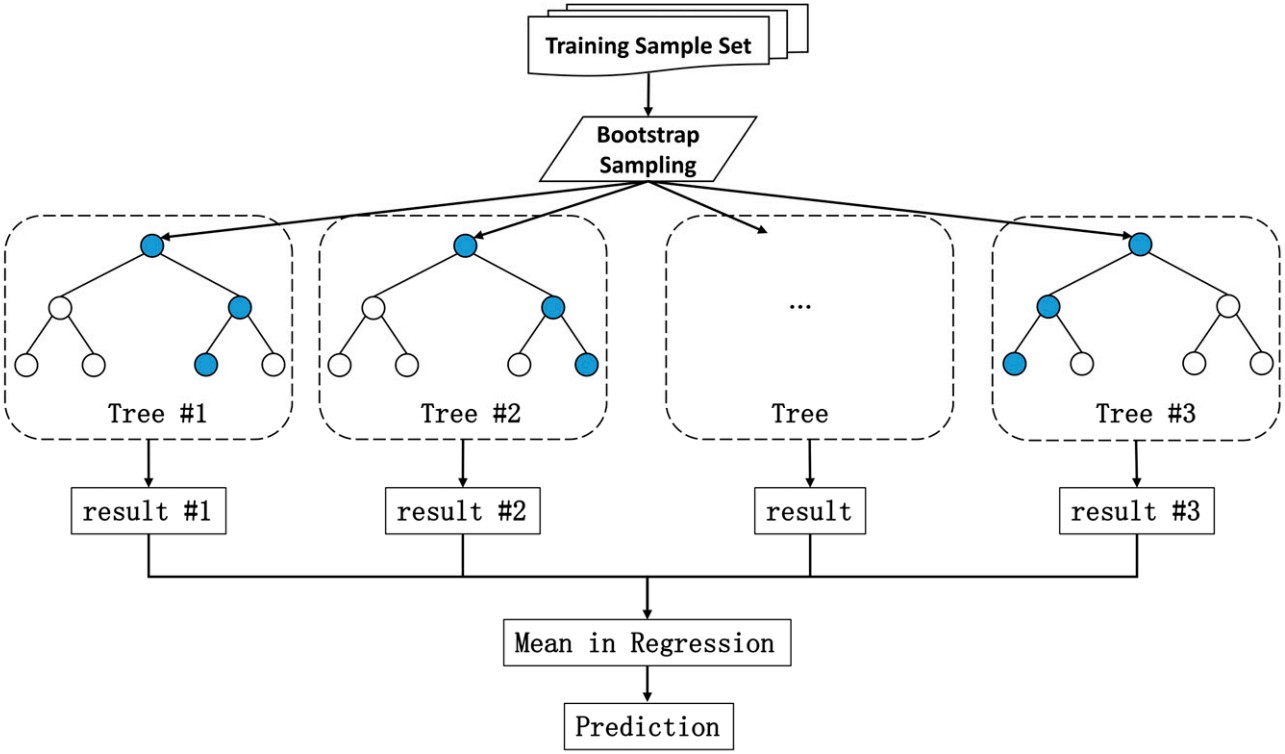

**Figure 3.** Random forest regression model construction process.

(1)    Using the Bootstrap sampling method with put-back, n samples are randomly selected from the original dataset, and the samples that are not drawn (Out of Bag, OBB) form the test set;

(2)    Construct n decision trees, select m features from the training sample data, choose the best feature to split, and keep splitting each tree until all training samples at that node belong to the same class;

(3)    Repeat both steps (1) and (2), and finally form the generated multiple classification trees into a random forest regression model;

(4)    Integrate all the generated decision trees for prediction to obtain the final prediction results.

Other atmospheric component profile sample data sources are mainly from ACE-FTS products. The amount of data is small for all the profile samples in the same grid to adopt the stratified mean method, according to each height layer, to find the mean value of the profile samples to obtain a standard profile to represent the atmospheric parameters in this grid. Equation (6) for the stratified mean method is as follows:

$$\bar{x} = \frac{\sum_{i=1}^{n} x_i}{n} \tag{6}$$

### 3. GRAP-Based Simulation Validation

This study employs the RFM atmospheric radiative transfer model to simulate the location and absorption intensity of absorption spectral lines for $CO_2$ and $CH_4$ in both

the full band and sensitive band ranges. The simulations select the Fengyun-3E HIRAS-II sample situated at mid- to low-latitudes (19.12°N, 98.65°E) in October. The parameters of the RFM were configured to match the surface temperature, surface reflectance, observation geometry, and spectral resolution (0.625 cm$^{-1}$) of the HIRAS-II transit moment sampling image element. The spectrum data from HIRAS-II was captured on 10 October 2022, at 11:40 AM, with a spatial resolution of 14 km. The file name associated with the data is FY3E_HIRAS_GRAN_L1_20221010_1140_014KM_V0.HDF.

Figure 4 presents the comparison between the spectrum data obtained from HIRAS-II and the simulated spectrum of the RFM in the three atmospheric models across the full band range of 650–2550 cm$^{-1}$. The results in Figure 4 indicate that the simulated spectrum of GRAP exhibits a much closer agreement with the measured spectrum from HIRAS-II.

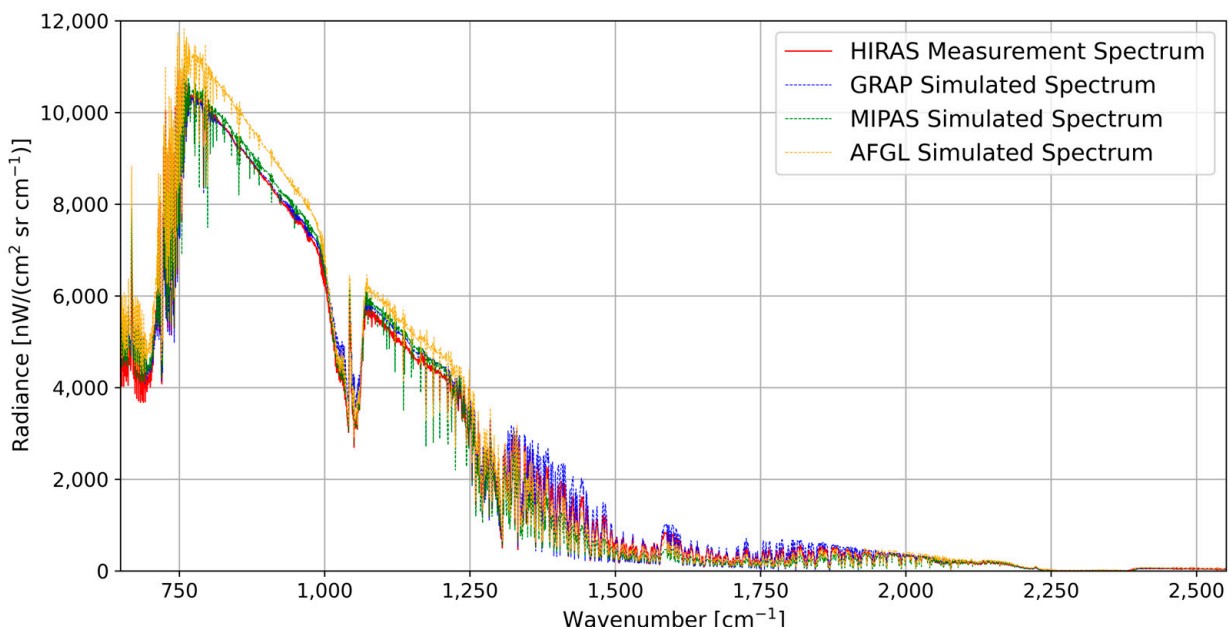

**Figure 4.** Comparison of the simulated spectrum and the measured spectrum of HIRAS-II in the 650–2550 cm$^{-1}$ band (full absorption band).

The absorption characteristics of different gases vary across different spectral bands. Figure 5 illustrates the spectrum within the range of 650–760 cm$^{-1}$, which corresponds to the strong absorption band of $CO_2$. This band is influenced by interfering gases such as $H_2O$, $N_2O$, $O_3$ and $HNO_3$. Similarly, Figure 6 presents the spectrum within the range of 1200–1400 cm$^{-1}$, which represents the strong absorption band of $CH_4$. This band is affected by interfering gases such as $H_2O$, $N_2O$, $CO_2$, $CF_4$ and $O_3$. By comparing the four spectrum curves in Figures 7 and 8, it is evident that the measured HIRAS spectrum (represented by the red solid line) closely aligns with the simulated GRAP spectrum (represented by the green dashed line). The deviations between the three simulated spectrum curves and the HIRAS-II measured spectrum curves are calculated. Figure 7 presents the GRAP simulated $CO_2$ absorption band spectrum within −10% to 12%, and Figure 8 presents the GRAP simulated $CH_4$ absorption band spectrum within −30% to 25%. The simulated spectrum of GRAP exhibits smaller deviations compared to the simulated spectrum of AFGL and MIPAS. This finding indicates that the atmospheric profile values employed in the RFM model within GRAP exhibit better consistency with the true values of the current atmospheric state.

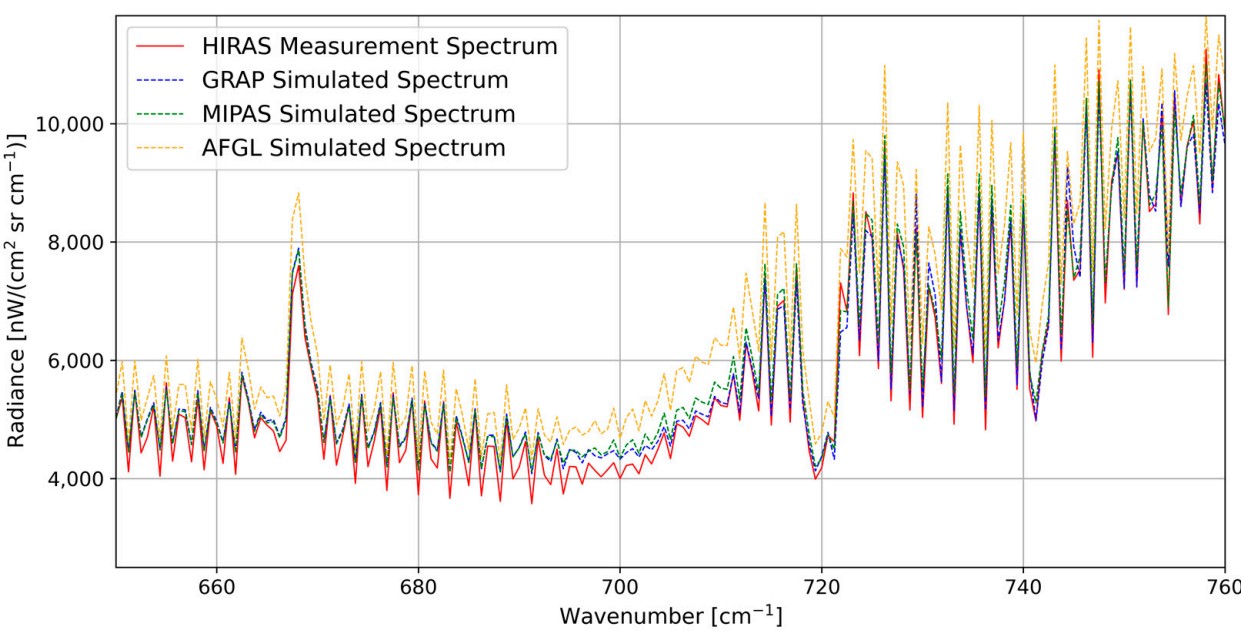

**Figure 5.** Comparison of the simulated spectrum and the measured spectrum of HIRAS-II in the 650–760 cm$^{-1}$ band (CO$_2$ absorption band).

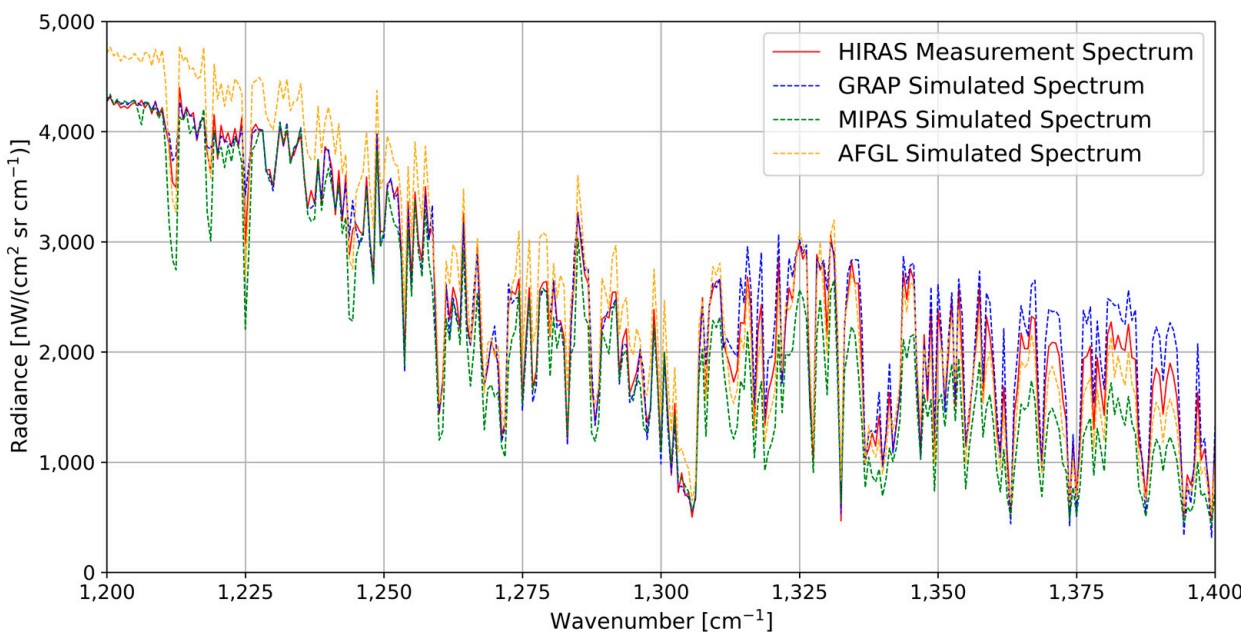

**Figure 6.** Comparison of the simulated spectrum and the measured spectrum of HIRAS-II in the 1200–1400 cm$^{-1}$ band (CH$_4$ absorption band).

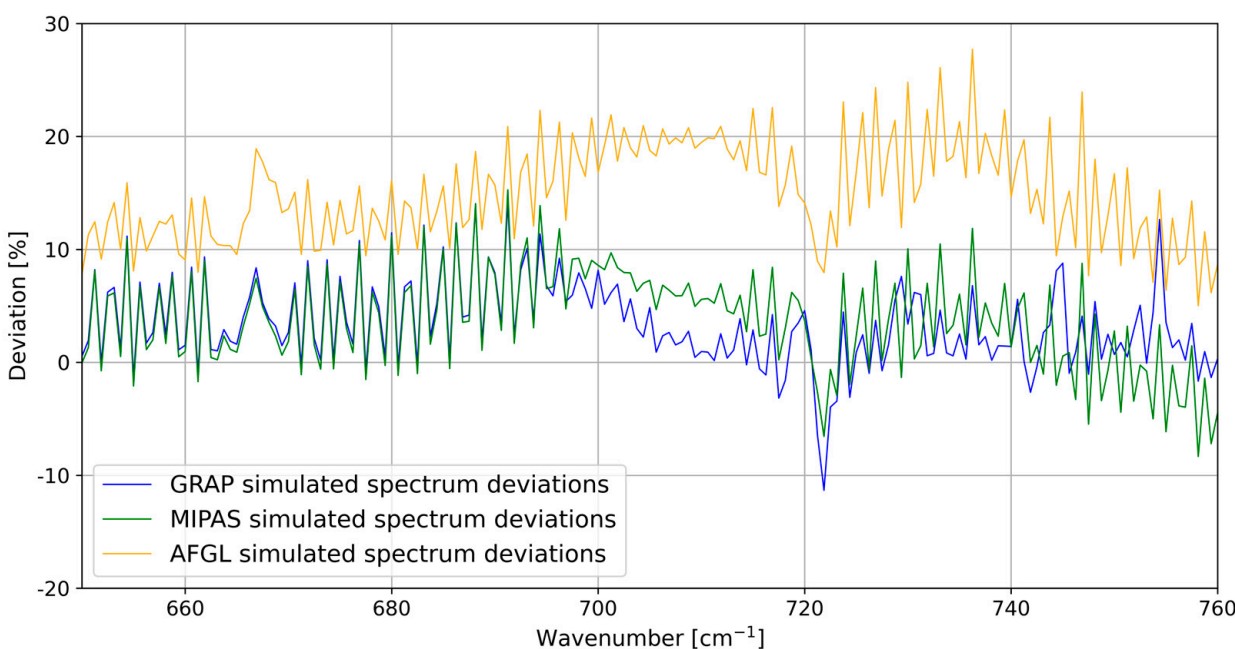

**Figure 7.** Deviation of the simulated and measured spectra of HIRAS-II in the 650–760 cm$^{-1}$ band.

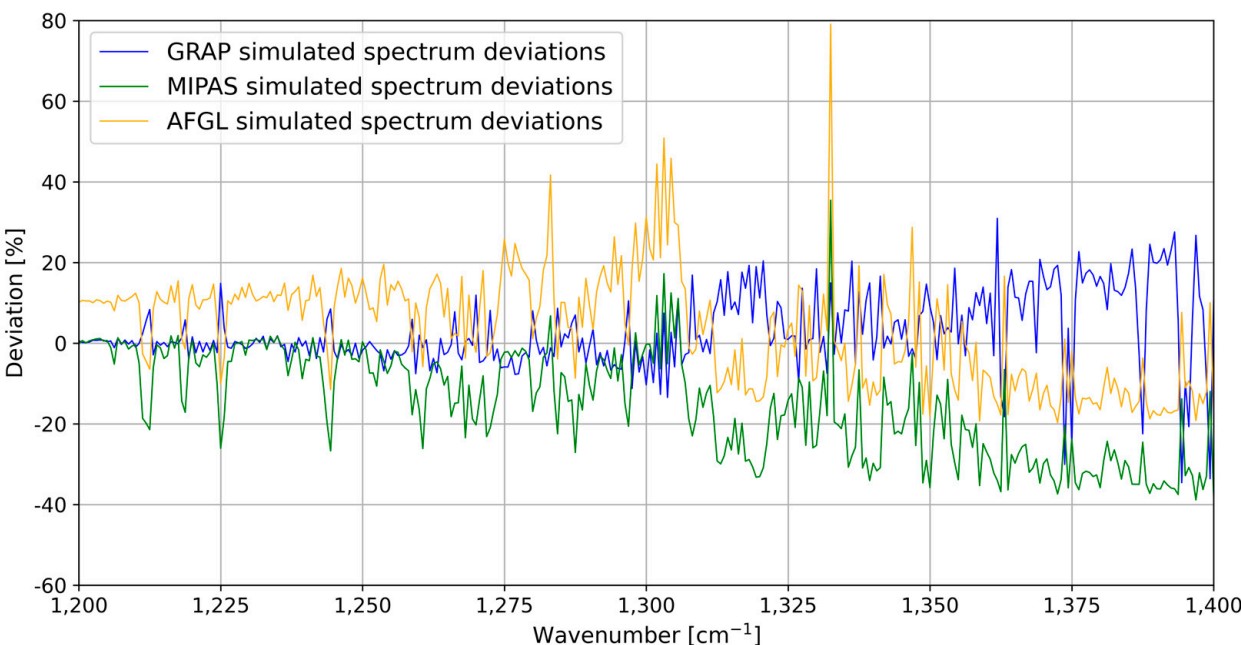

**Figure 8.** Deviation of the simulated and measured spectra of HIRAS-II in the 1200–1400 cm$^{-1}$ band.

## 4. Comparison of Reference Profiles and Discussion

This article compares and analyzes the profiles of $CO_2$, $CH_4$, $O_3$ and $N_2O$ in different latitude zones in the GRAP database, the AFGL database [22], the MIPAS database [31], the RTTOV database [32], and the profiles of $CH_4$, $O_3$ and $N_2O$ of the NDACC ground station in different latitude zones. It further selects and analyzes the temperature reference profiles of eight different grids in the Chinese region.

### 4.1. Comparison of Equatorial Reference Profiles

This study compares various atmospheric profiles in the equatorial climate zone, including the reference profiles of GRAP for the 0n0e grid in July, the equatorial reference profiles of the AFGL, the MIPAS equatorial reference profiles, the RTTOV reference profiles,

and the profiles measured in July 2021 at the Izaña ground station in Tenerife, Spain, which is located at the equator and affiliated with the Network for the Detection of Atmospheric Composition Change (NDACC).

The $CO_2$ profiles in Figure 9a exhibit overall consistency among the four databases. However, below 80 km altitude, the $CO_2$ profile values of AFGL are 330 ppmv, whereas the $CO_2$ profile values of MIPAS, RTTOV and GRAP are approximately 370 ppmv, 400 ppmv and 418 ppmv, respectively. There is a significant degree of difference between the four reference profiles. Between 80 and 120 km, the three reference $CO_2$ profiles of GRAP, MIPAS and AFGL experience a rapid decrease, whereas the $CO_2$ profile of GRAP remains higher than the other two profiles. Figure 9b presents the comparison of the $CH_4$ profiles, showing that the shape of the GRAP $CH_4$ profile resembles the other three profiles. However, its values are consistently higher from 0 to 120 km, peaking at approximately 1.95 ppmv in the troposphere. The primary focus of $CH_4$ is in the troposphere, and the comparison between the four profiles and the observed profiles from the NDACC ground station reveals the smallest difference between the GRAP profile and the NDACC observations in the troposphere. Figure 9c indicates minimal differences in $O_3$ profile values between GRAP and MIPAS, AFGL, NDACC, with the peak $O_3$ concentrations occurring at 28–32 km. In contrast, the differences between the RTTOV reference atmospheric profiles and the other three reference profiles are more pronounced. Figure 9d illustrates that the $N_2O$ profile values of GRAP, RTTOV and NDACC are slightly larger than those of AFGL and MIPAS within the 0–15 km range, while MIPAS values are larger within the 15–45 km range, with $N_2O$ concentration reaching approximately zero above 50 km.

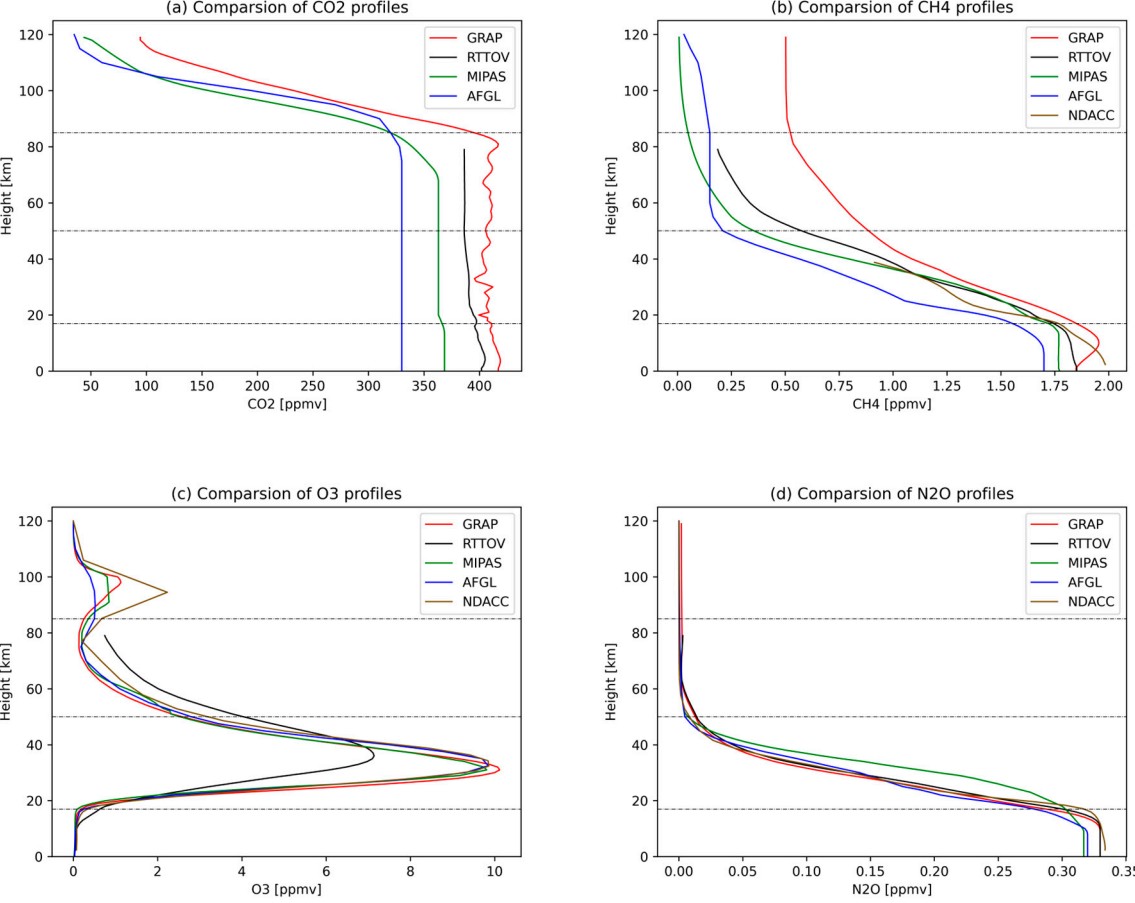

**Figure 9.** Comparison of standard equatorial profiles.

The four atmospheric profiles were utilized to calculate their corresponding total column density, and the resulting values are presented in Table 3. According to the *WMO Greenhouse Gas Bulletin (No. 18, 2022)* [33], published by the World Meteorological Organization (WMO), annual average global atmospheric concentrations of major greenhouse gases reached new highs in 2021. These are $415.7 \pm 0.2$ ppmv for $CO_2$, $1.908 \pm 0.002$ ppmv for $CH_4$ and $0.3345 \pm 0.0001$ ppmv for $N_2O$, which are 149%, 262% and 124% of pre-industrial (pre-1750) levels, respectively. The total column density measurements of atmospheric constituents at the equatorial Ascension Island ground station from the World Data Centre for Greenhouse Gases (WDCGG) were selected for validation, yielding values of 415.53 ppmv for $CO_2$, 1.868 ppmv for $CH_4$, and 0.334 ppmv for $N_2O$. In summary, the atmospheric parameter profile values of GRAP demonstrate closer agreement with the current atmospheric state compared to AFGL, RTTOV and MIPAS.

**Table 3.** Equatorial profiles correspond to total column density (units are ppmv).

| Atmospheric Composition | GRAP | RTTOV | MIPAS | AFGL | NDACC | WDCGG | WMO |
|---|---|---|---|---|---|---|---|
| $CO_2$ | 415.25 | 401.54 | 368.03 | 330 | - | 415.53 | 415.7 |
| $CH_4$ | 1.875 | 1.807 | 1.744 | 1.648 | 1.913 | 1.868 | 1.908 |
| $O_3$ | 0.369 | 0.301 | 0.304 | 0.346 | 0.374 | - | - |
| $N_2O$ | 0.323 | 0.319 | 0.311 | 0.307 | 0.326 | 0.334 | 0.3345 |

### 4.2. Comparison of Reference Northern Hemisphere Mid-Latitude Summer Profiles

This study compares atmospheric profiles for summer in the mid-latitude climatic zone. The selected profiles include reference profiles of GRAP for July located in the US region (40n90w) and China region (40n120e), mid-latitude summer reference profiles of the AFGL, mid-latitude daytime reference profiles of the MIPAS, the RTTOV reference profiles, and NDACC measured profiles for July 2021 at the Boulder ground station site in Boulder, CO, United States.

Figure 10a illustrates the comparison of $CO_2$ profiles, revealing the oscillation of $CO_2$ profile values of GRAP in the 18–75 km altitude range in both the China and US regions, with a maximum difference of 15 ppmv. However, above 75 km, the $CO_2$ values are lower in the US region compared to the China region. Figure 10b presents the comparison of $CH_4$ profiles. Between 0 and 20 km, there is a slight difference between the $CH_4$ profiles of GRAP in the US and China regions. The profile values of GRAP demonstrate closer agreement with the measured values at the NDACC ground station than the other three reference profiles. The $CH_4$ profile values of RTTOV significantly exceed those of the other three atmospheric models at 20–55 km. Above 55 km, the $CH_4$ profile of GRAP remains relatively constant and is notably higher than the other three atmospheric profiles. Figure 10c displays the comparison of $O_3$ profiles, indicating two peaks in the 25–55 km and 85–100 km ranges. The $O_3$ profiles of GRAP in the US and China regions differ by 0.2 ppmv, and the maximum difference among the six $O_3$ profile values is 2 ppmv. Figure 10d presents the comparison of $N_2O$ profiles, showing that the six profiles exhibit a similar trend with differences primarily located below 60 km. The $N_2O$ profile of GRAP exhibits the least difference from the measured profile at the NDACC ground station.

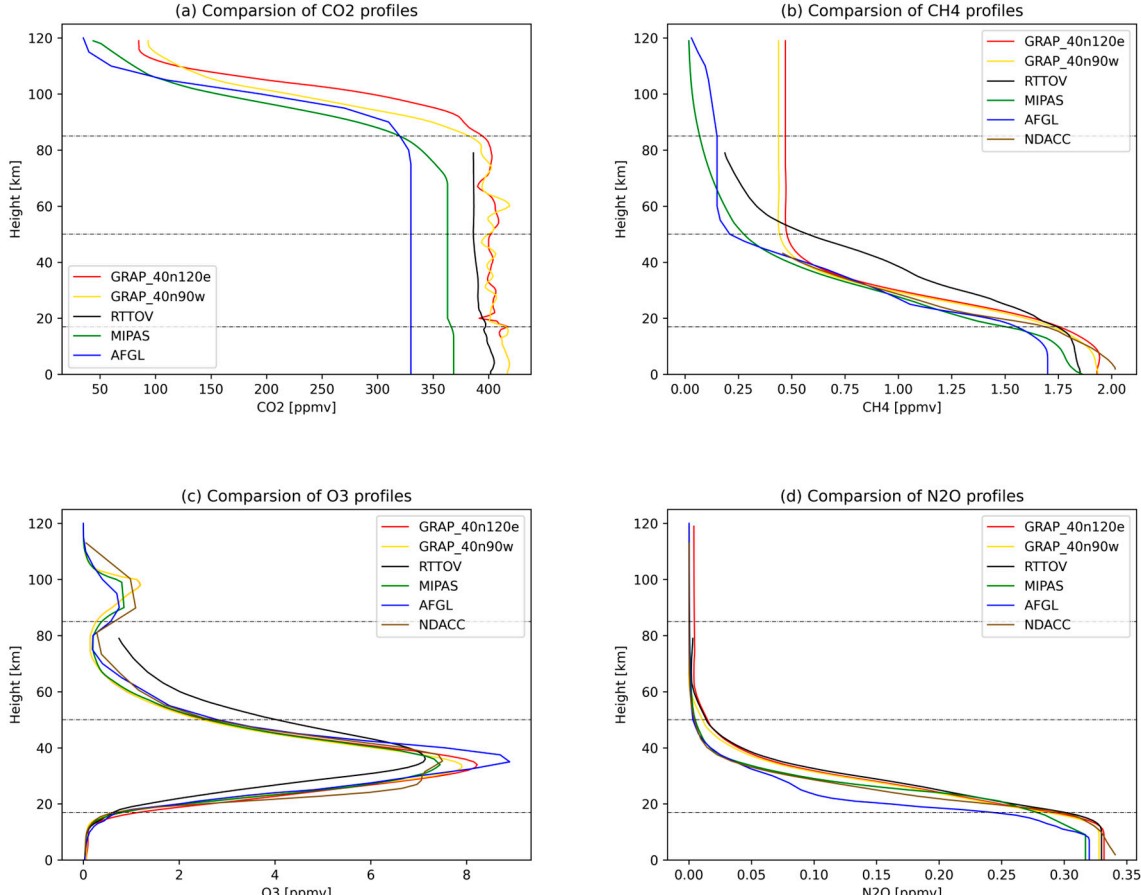

**Figure 10.** Comparison of standard summer profiles in the northern hemisphere at mid-latitudes.

The total column density values for the six atmospheric constituents are presented in Table 4. According to *the China Greenhouse Gas Bulletin No. 11 report* [34], observations from the China Meteorological Administration's Waliguan National Atmospheric Background Station in 2021 indicate annual average atmospheric concentrations of $CO_2$, $CH_4$ and $N_2O$ as $417.0 \pm 0.2$ ppmv, $1.965 \pm 0.0006$ ppmv, and $0.3351 \pm 0.0001$ ppmv, respectively. These values are comparable to the same period in the northern hemisphere mid-latitudes, although slightly higher than the global mean. It is important to note that, despite being in the same latitudinal zones, the atmospheric conditions can vary across different longitude zones. In terms of data source, the GRAP is more recent, ensuring that the atmospheric parameter profiles align more closely with current atmospheric conditions when compared to the other three reference profiles.

**Table 4.** Northern Hemisphere mid-latitude summer profiles correspond to total column density (units are ppmv).

| Atmospheric Composition | GRAP_40n120e | GRAP_40n90w | RTTOV | MIPAS | AFGL | NDACC | CGGB |
|---|---|---|---|---|---|---|---|
| $CO_2$ | 415.15 | 415.46 | 401.54 | 368.03 | 330 | - | 417 |
| $CH_4$ | 1.878 | 1.859 | 1.807 | 1.726 | 1.648 | 1.881 | 1.965 |
| $O_3$ | 0.451 | 0.378 | 0.301 | 0.375 | 0.392 | 0.469 | - |
| $N_2O$ | 0.328 | 0.327 | 0.319 | 0.304 | 0.297 | 0.322 | 0.3351 |

### 4.3. Comparison of Reference Polar Winter Profiles

This study compares reference profiles for the polar winter climatic zone. The selected profiles for comparison include the reference atmospheric profiles of the 90n30e grid of GRAP in January, the reference polar winter profiles of the AFGL, the reference polar

winter profiles of the MIPAS, the RTTOV reference profiles, and the profiles measured in March 2022 at the NDACC ground station in Ny Ålesund, Norway, which is located in the polar regions.

Figure 11a presents the comparison of the $CO_2$ reference profiles, which exhibit a similar trend to the equatorial and mid-latitude regions. Figure 11b presents the comparison of the $CH_4$ reference profiles. The $CH_4$ profile of GRAP exhibits higher values compared to the other three reference profiles, with a concentration peak at 15 km reaching nearly 2 ppmv. Furthermore, the GRAP values in the troposphere closely align with the measured profile values in the NDACC. Figure 11c presents the comparison of the $O_3$ profiles. The GRAP profile represents the lowest $O_3$ concentration, reaching as low as 4.7 ppmv, while the RTTOV profile exhibits the highest $O_3$ concentration, reaching up to 7.2 ppmv. The $O_3$ profile concentration from AFGL is in closer agreement with the measured profile from NDACC. Figure 11d illustrates the comparison of the $N_2O$ profiles. The trends in the five $N_2O$ profiles align closely with the equatorial and mid-latitude regions, with the $N_2O$ profile values of the GRAP and NDACC measured profiles being highly similar.

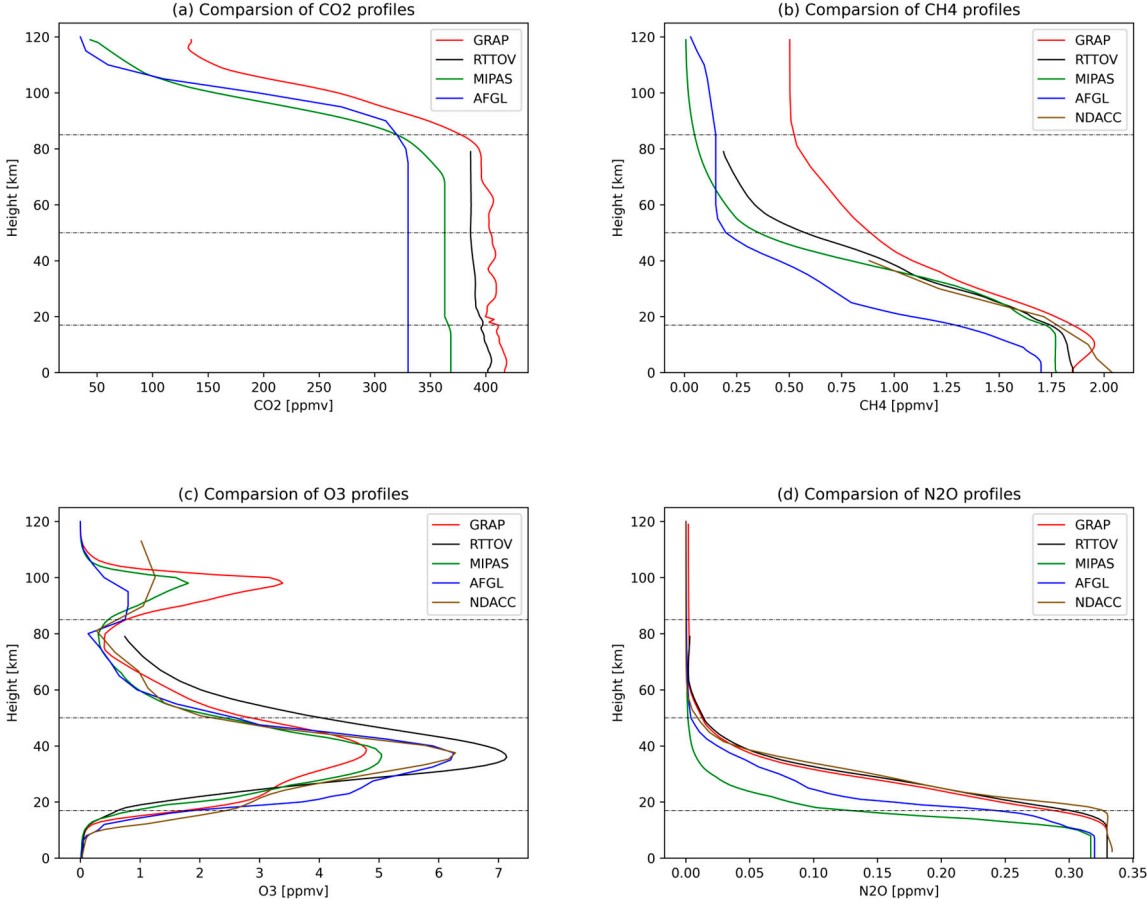

**Figure 11.** Comparison of standard polar winter profiles.

### 4.4. Comparison of Reference Atmospheric Temperature Profiles in China

Table 5 presents the characteristics of the selected Chinese regional reference atmospheric temperature profiles. Temporally, representative months for each season, namely January for winter, April for spring, July for summer, and October for autumn, were chosen in the Chinese region. Spatially, the profiles encompass various regions in China, including the northern, central, southern, northeastern, northwestern and southwestern parts of the country.

**Table 5.** Location of the Regional Reference Atmospheric Temperature Profile in China.

| Profile Number | Location Name | Latitude and Longitude | Grid Location |
| --- | --- | --- | --- |
| 1 | Harbin | 45.75N, 126.63E | 45n120e |
| 2 | Urumqi | 43.77N, 87.68E | 45n90e |
| 3 | Beijing | 39.92N, 116.42E | 40n120e |
| 4 | Shanghai | 34.50N, 121.43 | 35n120e |
| 5 | Lhasa | 29.60N, 91.00E | 30n90e |
| 6 | Kunming | 25.05N, 102.73E | 25n90e |
| 7 | Guangzhou | 23.13N, 113.27E | 25n120e |
| 8 | Sanya | 18.25N, 109.5E | 20n120e |

Figure 12 presents the comparison of reference temperature profiles for eight regions in China. Analysis of the figure reveals variations in surface temperatures (0 km) among all regions, with a maximum difference of 30 K occurring in winter, a minimum difference of only 15 K in summer, and approximately 20 K differences in both spring and autumn. In the troposphere, temperatures decrease with altitude, exhibiting the smallest variations in summer and a nearly parallel decreasing temperature profile. However, during spring, autumn and winter, a significant inflection point is observed around 10 km in Harbin, Urumqi, Beijing and Shanghai, where the rate of temperature decrease diminishes and even shows a rising trend. In the stratosphere, temperatures increase with altitude, displaying substantial differences between summer and winter. Harbin, Urumqi and Beijing exhibit significantly higher temperatures compared to the other five regions, while spring and autumn temperatures exhibit less disparity. In the mesosphere, temperatures decrease with altitude, reaching values of 160–190 K at the mesosphere's upper boundary. Finally, in the thermosphere, temperatures rise rapidly with altitude, peaking at 440 K.

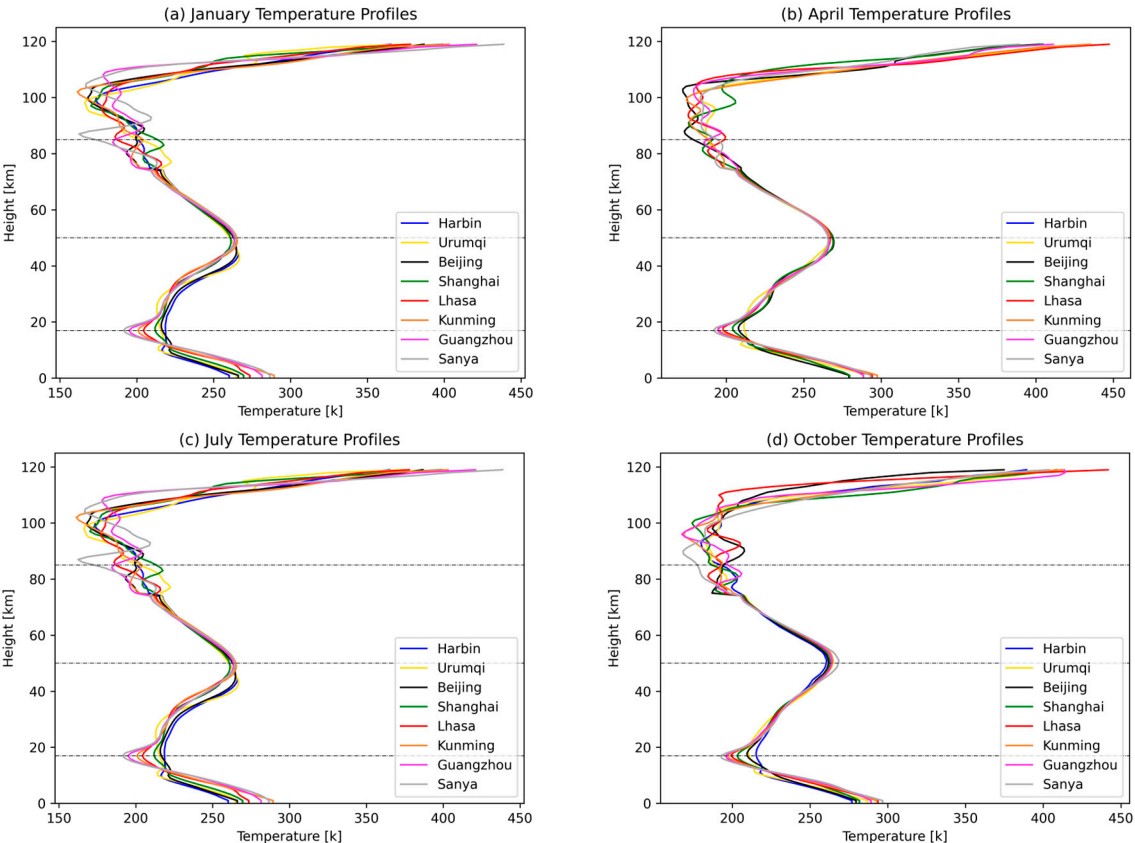

**Figure 12.** Comparison of regional atmospheric temperature standard profiles in China.

The analysis of temperature profile characteristics in various regions of China reveals that the temperature in the troposphere decreases as the latitude of the region increases.

During spring, autumn and winter, the troposphere's upper boundary decreases with latitude in eight regions, reaching approximately 15 km in Harbin, Urumqi and Beijing. In summer, the troposphere's upper boundary is around 18 km in all seven regions except Urumqi, where it stands at 16 km. The studies by Qi Chengli reveal significant seasonal and regional variations in the vertical distribution of temperature profiles in China [15,16]. Comparing the reference temperature profiles of GRAP for China with Qi Chengli's studies and the actual situation in China yields similar results, suggesting that latitudinal variations can lead to substantial disparities in tropospheric temperatures. Therefore, further subdivision of the atmospheric reference profile based on latitude holds considerable significance.

## 5. Conclusions

Most reference atmospheric profile databases commonly used have limited spatial and temporal resolution. These classification criteria fail to meet the research requirements for global regions with intricate topography and diverse climates. To address this, this study constructs the GRAP. The random forest regression model and the stratified mean method are adopted to process the ACE-FTS L2 products, the AIRS L2 products and the ERA5 reanalysis data. The data is divided into monthly intervals from January to December and spatially organized into 532 grids with dimensions of $5° × 30°$, creating a comprehensive global atmospheric profile reference database. Leading to the following conclusions:

(1) GRAP provides extensive coverage on a global scale, presenting a comprehensive composition of the atmosphere that accurately presents its current state.
(2) Four atmospheric reference profile databases were used as input parameters for the RFM radiative transfer model to simulate the FY-3E HIRAS-II absorption spectrum and compare it with the measured spectrum. The results illustrate that the spectral simulation of GRAP as an input parameter is a better fit for the measured HIRAS-II spectrum.
(3) The atmospheric profiles of the four reference atmospheric profile databases were compared to the measured atmospheric profiles from NDACC and the column total concentrations measured by WDCGG. The findings indicate substantial updates in the gas components of GRAP compared to the other three databases. Notably, the four greenhouse gases ($CO_2$, $CH_4$, $O_3$ and $N_2O$) of GRAP demonstrate better alignment with the current atmospheric conditions.
(4) Comparing the reference temperature profiles of GRAP for eight distinct regions of China reveals that these profiles effectively capture the climatic conditions. The fine spatial and temporal grids enable GRAP to achieve superior regional representativeness compared to previous reference atmospheric profile databases. Consequently, GRAP exhibits enhanced regional representation capabilities.

**Author Contributions:** Conceptualization, X.L.; Methodology, Y.G.; Software, Y.G.; Formal analysis, Y.G.; Investigation, X.Z., W.L. and W.F.; Data curation, Y.G.; Writing—original draft, Y.G.; Writing—review & editing, Y.G., X.L., T.C. and S.L.; Supervision, X.L.; Project administration, X.L.; Funding acquisition, X.L. All authors have read and agreed to the published version of the manuscript.

**Funding:** This research was funded by the National Key Research and Development Program of China under grant number 2022YFF0606400, the Basic Strengthening Program Technical Field Foundation, grant number 2023KJC-Y-0193, and Unit Commissioned Projects, grant number 2023KJC-Y-0032.

**Data Availability Statement:** Not applicable.

**Acknowledgments:** The AIRS Support Level 2 Version 7 products used in this research were downloaded from the NASA Goddard Space Flight Center Earth Science Data and Information Service, the ACE-FTS Level 2 Version 4.1 products were downloaded from the Atmospheric Chemistry Experiment Science Operations Centre, Department of Chemistry, University of Waterloo, the ERA5 reanalysis data were downloaded from the Copernicus Climate Change Service Climate Data Repository, and HIRAS-II L1 data from the National Centre for Meteorological Satellites (NCMMS) Feng Yun Satellite Remote Sensing Data Service Network, the NDACC ground station profiles data used

in this study were downloaded from the NDACC Rapid Delivery (RD) Data Access, for which we would like to express our sincere thanks!

**Conflicts of Interest:** The authors declare no conflict of interest. The funders had no role in the design of the study; in the collection, analyses, or interpretation of data; in the writing of the manuscript; or in the decision to publish the results.

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
