# Peer review of "Construction of the Global Reference Atmospheric Profile Database"

_remotesensing, doi:10.3390/rs15123006_

Round 1

Reviewer 1 Report

Based on ACE-FTS satellite data, AIRS satellite data, and ERA5 reanalysis data, this paper attempts to construct a Global Reference Atmospheric Profiles Dataset (GRAP) through the random forest method. The results show that GRAP has new data source time, high spatial and temporal resolution and better fits the current real atmospheric state. The paper is well organized. I suggest accept after minor revision.

1) Line 81: but spatial”-temporal” distribution? Because the atmospheric profiles represent the summer and winter.

2) Considering to move section 2 after section 3.2 and before section 3.2.1?

3) The font in Figure 2 is too small to read clearly.

Minor editing of English language required

Author Response

Dear reviewer:
      Thank you for your detailed comments. I have made changes to your comments. The attached are the specific changes I have made to your comments for your review.

Reviewer 2 Report

Utilizing random forest regression model and hierarchical mean algorithm, a short-term global grided profiles dataset for dozens of atmospheric parameters and compositions were constructed (GRAP for short), based on multiple types of satellite and reanalysis data. GRAP profiles of critical greenhouse gases at typical region and seasons were compared with other 3 atmospheric models, suggesting improvements.

Atmospheric profiles are quite important for improvements of radiative transfer models, ground-based remote sensing technologies and satellite observations. Apart from that, profiles for meteorological parameters and atmospheic compositons with high quality and better resolutions can deepen our understanding of the mechanisms and mutual influences of weather events, climate change issues and ecological feedbacks. It is of great importance for the science community to explore more potentials and produce more atmospheric profile products.

The manuscript is reasonably written and efforts were made to evaluate the results.

Major concerns:

2.1 The comparison and validation of the GRAP profile should be improved.

The most important characteristics of the profile are the stuctures of the distribution of the atmospheric parameter and their concentration ranges. The validation of the GRAP profiles were insufficient in this manuscript.

For one example, in Sect.5.1, the CO2 profile of equatorial region from GRAP was considered to be more ‘closer’ to real CO2 state by comparing with annual aveage global concentration from WMO greenhouse gases bulletin. For one thing, this annual aveage global concentration of CO2 was calculated from 32 global WMO GAW stations’ annual average concentration of CO2 across the whole world, which is not suitable for validating the profile’s concentration range at equatorial region in July. The authors are suggested to use the CO2 concentrations at equatorial region in summer period for comparison (data can be found in data center of WMO, like WDCGG (World Data Centre for Greenhouse Gases) (kishou.go.jp)).

For another thing, this annual average concentration is just surface concentration. A ‘closer’ concentration at surface doesnot mean the accuracy for the whole profile. As a result, the authors are suggested to use real profile data for this validation. There are aircraft measurement or Aircore data for CO2 and CH4, and ozone sounding data for ozone profiles.

 2.2 The conclusion of Sect 5.4 has not been supported by the analysis. The GRAP temerature profiles of different cities in four seasons were compared to each other, withour validation to real sounding data or existed studies. The conclusion ‘GTAP is a better representation of the state of the atmosphere in the Chinese region’ lacks research fundamentation.

 Specific comments

1)All the figures in the manuscript need to be replaced with higher resolutions. Font size of lables in the figures should be larger to make them more clear to readers.

2)Please give numbers or labels to each sub-figure, like (a)(b). At the mean time, the labels of subfigures are suggested locating at the up left corner of the sub-plots. Please revise the Figure 2, Figure 9 ~12.

3)line 353-355: rewrite the sentence, it is confusing. The authors probably mean the difference of peak values of ozone profiles from these models.

Generally speaking, the English writing is fine. Editing for certain sentences are needed, which can be found in Comments and Suggestions. 

Author Response

(The authors gave the same response as above.)

Round 2

Reviewer 2 Report

1. The comparison and validation of the GRAP profile have been improved.

2. Figure 2: Please give numbers or labels to each sub-figure, like (a)(b).

No further comment.

Author Response

Dear reviewer:
      Thank you for your detailed comments. I have revised to your comments. The attached are the specific changes I have made to your comments for your review.
